# The effectiveness of a two-layer neural network for recommendations

**Oleg Rybakov[1], Vijai Mohan[1], Avishkar Misra[2], Scott LeGrand[1], Rejith Joseph[1], Kiuk Chung[1], Siddharth Singh[1], Qian You[3], Eric Nalisnick[4], Leo Dirac[1], Runfei Luo[5]**
[1]Amazon.com, [2]Think Big Analytics, [3]Snap Inc, [4]UC Irvine, [5]UC Santa Barbara

## Abstract

We present a personalized recommender system using neural network for recommending products, such as eBooks, audio-books, Mobile Apps, Video and Music. It produces recommendations based on customer's implicit feedback history such as purchases, listens or watches. Our key contribution is to formulate recommendation problem as a model that encodes historical behavior to predict the future behavior using *soft* data split, combining predictor and auto-encoder models. We introduce convolutional layer for learning the importance (*time decay*) of the purchases depending on their purchase date and demonstrate that the shape of the *time decay* function can be well approximated by a parametrical function. We present offline experimental results showing that neural networks with two hidden layers can capture seasonality changes, and at the same time outperform other modeling techniques, including our recommender in production. Most importantly, we demonstrate that our model can be scaled to all digital categories, and we observe significant improvements in an online A/B test. We also discuss key enhancements to the neural network model and describe our production pipeline. Finally we open-sourced our deep learning library which supports multi-gpu model parallel training. This is an important feature in building neural network based recommenders with large dimensionality of input and output data.

## 1 Introduction

Recently, deep learning based recommender systems gained significant attention by outperforming conventional approaches (Zhang et al., 2017). It shows promising results on products like videos (Covington et al., 2016), mobile apps (Cheng et al., 2016), music (Van den Oord et al., 2013) etc.

In the papers mentioned above, we noticed that NN based recommenders are different for each product category (videos, music, mobile apps), requiring unique feature extraction methods and NN topologies. All of these challenges makes it harder to scale over different product categories. In this paper we are exploring effectiveness of a multilayer neural network (NN) for personalized recommendations of products which were never purchased before by a customer. The simplicity of this approach allows us to scale it on various categories of Amazon catalog in production. We focus on improving accuracy of the neural network based personalized recommender.

It is noticed in (Covington et al., 2016) that accuracy of NN depends on how the problem is formulated. They found that NN performs better when it is trained to predict the user's next purchase, rather than a set of randomly held-out purchases. We use the same idea, but on top of this, we propose to train NN model to predict not only future purchase, but all future purchases in the certain time (for example in the next week).

Capturing temporal popularity (trendiness) also called seasonal changes of consumption pattern is a challenging and important problem in recommender systems (Wu et al., 2017), which can impact the accuracy of the model over time. In (Wu et al., 2017; 2016; Ko et al., 2016) authors propose methods to capture seasonality changes using sequence modeling. Another approach (Song et al., 2016) models both long-term static and short-term temporal user preferences. In both cases they use different versions of recurrent neural networks. In this paper we propose to combine *predictor* model (which can captures short term preferences and recommend products which are currently popular)

with *auto-encoder* model (which can capture static customer preferences and recommend products which were popular at any time in the past) using feed forward NN. These models are combined by training them jointly. We re-train NN model every day to learn new popular products and changes in customer preferences. Even though our model is simpler then RNN, we show that it captures seasonality changes well.

Improving NN based recommender is important problem, for example in (Covington et al., 2016) authors observed that adding features and depth significantly improves precision on holdout data from YouTube catalog. In (Cheng et al., 2016) authors show that wide and deep NN with multiple features can improve performance of the neural network on mobile apps. So both methods (Cheng et al., 2016) and (Covington et al., 2016) require different production pipelines for different data sets: video and mobile apps. In this paper we use only purchase history and focus on improvements of NN accuracy by applying different splits of the training data. It simplifies the production pipeline and allows us to reuse it on all digital categories: video, eBooks, audio-books, mobile apps, and music.

Another way of improving recommender system is *time decay*, which was introduced by (Xia et al., 2010) for collaborative filtering. We also apply it on input data for the neural network based recommender and observe positive impact on accuracy metrics. Our contribution is to use convolutional layer (LeCun & Bengio, 1998) for estimating the shape of *time decay* function. Convolutional layers are used in existing recommender systems, but it is applied for different purposes, for example in (Hsu et al., 2016) convolutional layer is used for learning local relation between adjacent songs, in (Zheng et al., 2017; Kim et al., 2016) it is used for text feature extraction and in (Van den Oord et al., 2013) it is used for extracting features from audio signal.

There can be millions of products in the catalog and it is a hard problem to run NN based recommender in production with such amount of items (Covington et al., 2016). Both (Covington et al., 2016) and (Cheng et al., 2016) are splitting the problem into candidate generation and ranking. Candidate generation retrieves a small subset of products from a large corpus. These candidates should be relevant to customer interest. Ranking does a fine-level scoring of the candidates and in addition to consumption history it can use more features (context, impression, etc). Another way of scaling this problem is to learn similarity between products using DSSM approach (Elkahky et al., 2015) which is relying on content features. In this paper we focused on training end-to-end one neural network which is using only purchases events. On one hand it simplifies the production pipeline, because there is no splitting into candidate generation and ranking models and there is no special feature extraction step. But on the other hand we have to deal with large dimensionality of input features and labels. To solve this problem we use multi-GPU model parallelization, implemented by our team in DSSTNE library (10). It allows us to re-train large NN models every day and produce fresh recommendations for our customers.

In this paper, we are focused on modeling consumption patterns in digital products (For example, recommending movies to customers based on the movies already purchased). Depending on the domain, we also exclude movies that were already purchased by the customer while computing offline metrics as well as recommending online. We present different methods of splitting the training data and observed that it can improve NN based recommender accuracy metrics. Techniques like the one presented here feed into recommendation technology deployed at Amazon.

The rest of the paper is organized as follows. Section 2 introduces offline metrics used for algorithm evaluation. Section 3 details our NN model development procedure, including how different methods are compared. Section 4 provides extensive offline evaluation results, in conjunction with model property exploration. Section 5 presents how to run NN model in production and describes on-line A/B test. Finally, section 6 presents our conclusions.

## 2 OFFLINE EVALUATION METHODOLOGY

There are many different metrics focusing on specific properties of the recommendation algorithm (6, 2014). Among all, root mean square error (RMSE) is the most popular one ((Qu et al., 2016), (Sedhain et al., 2015)). It requires explicit feedbacks (ratings). Nevertheless, in many practical applications recommender systems need to be centered on implicit feedbacks (Hu et al., 2008). Implicit information like clicks and purchases are normally tracked automatically, customers do not

need to explicitly express their attitude, therefore are easier to collect. In the scenario of predicting future purchase from implicit feedback data, we use two metrics throughout evaluations in this paper: Precision at K and Product Converted Coverage (PCC) at K.

Precision at K is the accuracy of the predicted recommendations with respect to the actual purchases:

$$Precision@K = \frac{1}{C} \sum_{c=0}^{C-1} \frac{|\{Rec_c\} \cap \{T_c\}|}{K},$$ (1)

where $K$ is the position/rank of a recommendation, $c$ is the customer index, $Rec_c$ is top $K$ recommended items for customer $c$, $T_c$ is actual consumptions for customer $c$ represented as the set of items the customer purchased in the evaluation period (where interaction can be purchases, watches, listens), $|Rec|$ is the number of items in set $Rec$, $Rec \cap T$ is the intersection between sets $Rec$ and $T$, and $C$ is the number of customers.

While having high precision is necessary, it is not sufficient. A personalized recommender should also recommend diverse set of items (Adomavicius & Kwon, 2012). For example, if precision is high with no diversity, then recommendations looks like a *hall of mirrors* showing only products in a single topic. Therefore, to guarantee the diversity of recommendations, we use products converted coverage at K. It captures the number of unique products being recommended at top K and at the same time purchased:

$$PCC@K = \frac{1}{P} | \cup_c^{C-1} (\{Rec_c\} \cap \{T_c\})|,$$ (2)

where $\cup_c^{C-1}(X_c)$ represents union of sets $X_0, X_1 \ldots X_{C-1}$, $P$ is total number of products.

Using held-out labels to measure a recommender's efficacy is leaking future purchase information (Covington et al., 2016). Consequently, there exists the risk of having inconsistent performance between offline and online evaluation. In order to reduce this gap and emulate real production environment, the test metrics in this paper are measured on future purchases instead of held out data.

## 3   OFFLINE MODEL DEVELOPMENT

### 3.1   DATA PRE-PROCESSING

Apart from metric selection, one general issue related to a recommender system is how to reduce the gap between offline and online A/B test results. With the intention of using historical consumptions for training and future purchases for testing, we split data as shown in Figure 1. Users' consumption history is divided into past and future parts by pre-selected date ($dyz$), after which offline accuracy metrics are measured on future part. Each purchase event of a particular customer is represented by one-hot encoding in vectors: $X, Y, Z$, as shown in Figure 1, where: $X$ – training data input; $Y$ – training data output; $XY$ – testing data input (concatenation of $X$ and $Y$); $Z$ – testing data output; $dx, dxy$ – minimum and maximum dates in training data input $X$; $dyz, dz$ – minimum and maximum dates of purchase in testing data output $Z$. They are selected so that $dz - dyz$ is between 1 week and 2 weeks.

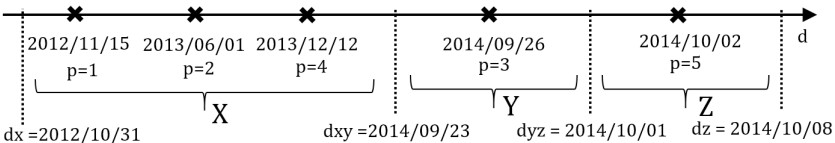

Figure 1: Purchase history data split. Consumption history is divided into three pieces $X$, $Y$ and $Z$, by date split $dx$, $dxy$, $dyz$ and $dz$. $p$ is index of the product purchased at YYYY/MM/DD. For example, $p = 1$ – product 1 purchased at 2012/11/15, $p = 2$ – product 2 purchased at 2013/06/21.

In practice, $dxy$ is selected so that $dyz - dxy$ is between 1 week and 3 months. By doing this, neural network will capture seasonality change. $dx$ is selected so that $dxy - dx$ is less than 2 years. During

training we consider $X$ as past purchases and our goal is to predict future purchases $Y$. During testing we concatenate $X$ and $Y$ data and $XY$ is used to predict $Z$.

Purchase history is inherently noisy (Hu et al., 2008). It has different kinds of noises. For example, customers with a small number of purchases (which may not be reflected of their true taste due to the small sample), customers with a lot of purchases (which might indicate a company or bulk account). To deal with this, we set the minimum and maximum number of purchases and exclude customers outside of these limits.

We apply the same data split (Figure 1) for extracting validation data sets which are used for hyper parameters optimization. Validation data for $X, Y, Z$ are extracted using date thresholds shifted by one month $dx - month$, $dxy - month$, $dyz - month$, $dz - month$. Then we train models with different hyper parameters on $X$ and $Y$ and pick the one which has highest accuracy on $Z$. Date thresholds $dx$ and $dxy$ can be treated as hyper-parameters too which are optimized to achieve maximum accuracy on output data $Z$.

During offline evaluation, we limit number of products to order of $10^5$ and reduced number of testing samples to order of $10^6$. It is worth noting that the purchase data has two dominating properties. First, the distribution is long-tailed, more than 90% of purchases are covered by less than 20% of products. This fact allows us to reduce dimensionality of recommended products by four times with minimum influence on precision, but PCC can be impacted. Second, the purchase history is sparse. On average customers have purchased less then 1% of the all products in the input data $X$. For efficient model training, we use sparse matrix multiplication in our production pipeline.

## 3.2 NEURAL NETWORK BASED RECOMMENDER – PREDICTOR MODEL

It is important to learn what customers are interested in buying *now*. For example, given one customer's purchase history in the last two years, we aim to predict their next purchases in the next week. We use NN based approaches because they are showing promising results (Covington et al., 2016; Cheng et al., 2016).

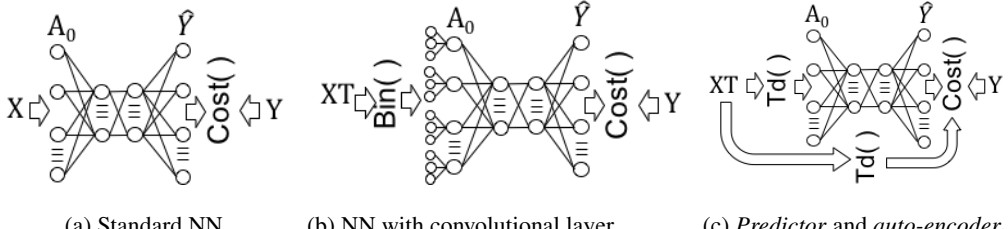

(a) Standard NN      (b) NN with convolutional layer      (c) *Predictor* and *auto-encoder*

Figure 2: Neural Network topologies

The recommender model for $L$ layers is defined by the function at each layer, and the loss function, as shown in Figure 2a and illustrated below using the equations corresponding to each layer:

$$A_0 = X + \xi; \quad A_j = Drop(Act(W_j \cdot A_{j-1} + B_j)); \quad \hat{Y} = Sigm(W_L \cdot A_{L-1} + B_L);$$

$$loss = Cost(Y, \hat{Y}) + \lambda \sum_{j=1}^{L} |W_j|^2 + \beta \sum_{j=1}^{L-1} KL(\rho, \rho'_j); \quad (3)$$

where weighted cross-entropy is used as the cost function:

$$Cost(Y, \hat{Y}) = -\sum_{p=0}^{P-1} (w_1 \cdot y_p \cdot log(\hat{y}_p) + w_0 \cdot (1 - y_p) \cdot log(1 - \hat{y}_p))$$

with, $X, Y$ – training input and output data in Figure 1; $X, Y \in \{0, 1\}^P$, $P$ – number of recommended products; $\hat{Y}$ – output scores of the NN; $L$ – number of layers in the NN; $W_j, B_j$ – weight matrix and bias to be learnt. $j = 1, \ldots, L$; $A_j$ is the activation at layer $j$; $Act(*)$ –

the nonlinear activation function (Relu, Sigmoid, (Nair & Hinton, 2010) etc.); $Drop(*)$ – drop out function (Srivastava et al., 2014); $\xi$ – noise added to input data, for increasing robustness of the model (Vincent et al., 2008); $w_1, w_0$ – constant weights for purchased and non-purchased products respectively, introduced for balancing number of purchased vs non-purchased products; $KL(\rho, \rho'_j) = \rho \cdot log(\frac{\rho}{\rho'_j}) + (1 - \rho) \cdot log\frac{1-\rho}{1-\rho'_j}$ – penalty function (Ng, 2011) on hidden and output nodes to force the network to only activate some percentage of the hidden and output nodes; $\rho$ – sparsity parameter; $\rho'_j$ – average activation of hidden unit $j$ (Ng, 2011).

Parameters of the noise $\xi$, drop-out function, number of layers $L$, set of activation function $Act(*)$, weights $w_1, w_0, \lambda$ and $\beta$ are hyper-parameters of the neural network model (3), which can be tuned via hyper-parameter search on validation data. We also explore learning rate, momentum, mini batch size, number of epochs, type of the optimizer (SGD, Nesterov (Nesterov, 1983), etc). We tested deeper networks on our data sets but did not observe significant improvements. In all the experiments mentioned in this paper, our models used neural networks with two hidden layers.

During evaluation we feed $XY$ data to the NN model and produce output scores $\hat{Y}$. Then these scores are sorted and the top K products are returned as recommendations. Before sorting, all previous purchases (products belonging to data $XY$) of the selected customer are removed from the recommendations, so that only new products are recommended. During evaluation, these recommendations are compared with data $Z$ for accuracy calculation.

The data split method used for training model (3) via the multi-label classification is shown in Figure 1. We call it *predictor* model because it predicts future purchases $Y$ given past purchases $X$. This model can produce personalized recommendations of products which were popular during the last week. As a result we can expect high precision of predicting future purchases, but PCC will be constrained by the products which were purchased in the last week ($Y$) only.

Gradient vanishing can be a problem in deep predictor model, so we use ReLU activation function to mitigate it on some categories of data. With increase of the depth (number of hidden layer) of predictor model, accuracy metrics can degrade significantly (vanishing gradient is one of the reason of such effect). That is why we measured the impact of the NN depth on Precision@1, and observed that with increasing the NN depth, Precision@1 is going down as follow (even with ReLU):

| Depth | 1 | 2 | 3 | 4 | 5 | 6 |
|---|---|---|---|---|---|---|
| Precision@1 | 0.072 | 0.072 | 0.07 | 0.068 | 0.067 | 0.065 |

One of the method of mitigating the accuracy degradation (due to depth of NN) is residual neural networks (He et al., 2015). We explored residual NN with predictor model on our data sets, and observed that it mitigates vanishing gradient effect, so that Precision@1 stayed the same regardless of the depth of the NN: around 0.072. But it does not improve accuracy metrics in comparison with two layers NN. That is why we picked neural network model with number of hidden layer no more than 2. Above experiment was done on AIV data sets (which is also used below in sections 3.3, 3.4, 4.1, 4.3, 4.4, 4.5).

### 3.3 OFFLINE COMPARISON OF PREDICTOR MODEL WITH OTHER APPROACHES

We check the effectiveness of *predictor* model by comparing it against the following models [1] on AIV datasets:
**FISM** (Kabbur et al., 2013) is based on item-to-item similarity matrix factorization, in which two low dimensional latent factor matrices are learned. Bayesian Personalized Ranking (BPR) (Rendle et al., 2009) loss with negative sampling is used. The model is tuned for different values of rank $k$ and learning rate $l$.
**Fossil** (He & McAuley, 2016) combines similarity matrix factorization which utilizes Markov chains to capture temporal dynamics. In addition to rank and learning rate, we also tune for $L_2$ regularizer and order of the Markov chains.
**LSTM** is commonly used for dealing with sequence of items (Graves, 2013). We use the unidirectional LSTM with a softmax output layer to choose top $k$ recommendations, as implemented in

---

[1]FastXML is implemented by `https://github.com/Refefer/fastxml`, all the rest is implemented by `https://github.com/rdevooght/sequence-based-recommendations` (Devooght & Bersini, 2016)

(Devooght & Bersini, 2016). Best result is selected upon different layer sizes $\{32, 64\}$, batch sizes $\{16, 32\}$ and size of an extra embedding layer before the recurrent layer $\{0, 32, 128\}$.

**GRU** Similar to LSTM, GRU is also a gated RNN in which information flows are controlled, but with a different gated recurrent unit (Hidasi et al., 2015). The model is tuned in same manner as LSTM.

**FastXML** Our recommender system can be formulated as an extreme multi-label classification problem. Therefore, we compare our approach with Fast and Accurate Tree Extreme Multi-label Classifier (Prabhu & Varma, 2014).

On Figure 3, we see that our model (3) has the best Precision, and our production CF has the best PCC. The next best model is FastXML. In the rest of this paper, we will be focusing on improving the NN model in terms of Precision and PCC.

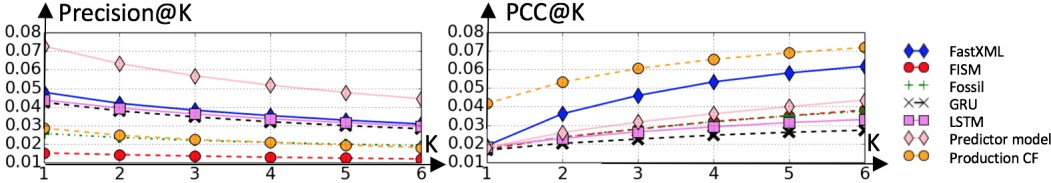

Figure 3: Accuracy metrics of different approaches on AIV purchase history

LSTM is well applied on sequences like text, speech etc. These sequences has *strong* grammatical rules, which are well captured by LSTM. We explain lower accuracy of LSTM (On Figure 3 ) by our data properties (or lack of *strong* grammatical rules in sequences of purchases in our data). For example on ebooks data, if one customer buy books in order: *Harry Potter*, *Golden Compass*, *Inkheart*, another customer can buy these books in different order: *Inkheart*, *Harry Potter*, *Golden Compass* and another one in different order, etc. So these purchases can be done in any order and *long* term dependencies can be noisy. Another important properties of our data(video, ebooks) is the popularity of the recommended products at particular date. Our approach (predictor model) is modeling these properties by re-training the model every day and predicting the next purchases which are popular in the current week, whereas LSTM is recommending only next purchases (which are not necessary popular at current week). We can expect better performance of LSTM on other categories of products (where order of purchases is more important), for example probability of buying a game for a cell phone after purchasing a cell phone is higher than probability of buying these products in reversed order.

### 3.4 LEARNING TIME DECAY FUNCTION WITH CONVOLUTIONAL LAYER

One of the ways to improve a recommender is by using *time decay* function (Xia et al., 2010), so that the importance of recent purchases can be increased. There are multiple parametrical functions that can be used for this purpose but we do not know which one of them is the best. That is why we propose to use a NN with convolutional layer that can learn the shape of the time decay function. NN model with convolutional layer is described below and shown on Figure 2b.

Date of purchase $XT$ is processed by $Bin()$ function, it converts purchase date into 32 binary encodings. For example, recent purchase will be represented as $[1, 0 \ldots, 0]$, whereas purchase which was done 2 years ago will be encoded as $[0 \ldots, 0, 1]$. Two years of purchase events are divided into 32 buckets, so that 22 days belongs to one bucket. All binary encoded data are concatenated into one vector $V$, which will be used by one dimensional convolutional layer (with stride 32).

$$V = Bin(XT);$$

$$A_0 = Conv(V) = Act(\sum_{i=0}^{31}(V_{i+p \cdot 32} \cdot wf_i + wb)), \tag{4}$$

where $wf$ is the one dimensional convolutional kernel (with size 32), and $wb$ is the bias for $p = 0 \ldots P-1$, where $P$ is the total number of products. The remaining functions are the same with (3).

To see how convolutional layer can explore the shape of *time decay*, we use a parametric function as a baseline. It is defined as:

$$Td(d, decay) = \begin{cases} 1/(1 + \frac{ref - d}{decay}), & d < ref \\ 0, & else \end{cases} \tag{5}$$

where $d$ – input day; $decay$ – time decay parameter measured in days; $ref$ – reference day, which is equal to $dxy$ during training and $dyz$ while generating recommendations.

We trained three models, the first model was a NN *predictor* defined by (3), the second used a convolutional layer defined by (4) and shown in Figure 2b, and the third one took dates of input purchases decayed by (5) and then fed this data to the input of model (3). The time decay parameter of the third model was learnt using hyper-parameter optimization on validation data, as described in section 3.1. The optimal value for this parameter was found to be $40$. These models were trained on AIV data sets with different date splits: October data set ($dx = 2012/10/31$, $dxy = 2014/09/23, dyz = 2014/10/01, dz = 2014/10/08$) and Christmas data set ($dx = 2011/10/31, dxy = 2013/12/15, dyz = 2013/12/22, dz = 2013/12/29$). We selected Christmas week to see that *time decay* function with convolutional layer will have a spike at Christmas week one year in the past.

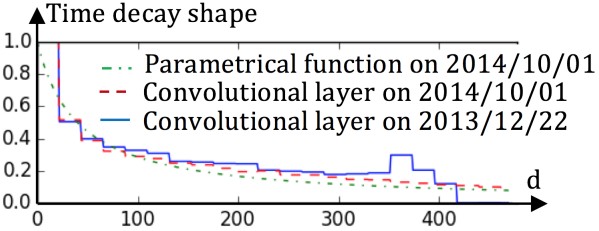

Figure 4: Shape of time decay function (4), (5). Parameter $d$ belongs to two years range: $0, \ldots, 365 \times 2$.

The shape of *time decay* function learnt by convolutional layer is shown on Figure 4. We observed that the function modeled on October data (date split $dyz = 2014/10/01$) is going down with increasing $d$ (red dashed line). So it can be well approximated by the parametrical function (5) (green dashed line).

For modeling seasonal activities like movies popular during Christmas every year, one would expect movies that were popular last Christmas week will be popular this Christmas week as well. In our experiment, as we expect, the convolutional layer learns a different shape of the time decay function for the Christmas week ($dyz = 2013/12/22$, blue solid curve). It is demonstrated by a local spike observed during that time, highlighting the popularity of Christmas movies at the same time of the year. This spike is located at Christmas week of the previous year.

Table 1 shows the results that compare *predictor* model with and without time decay. Time decay on input data slightly improves precision, but PCC is reduced. We demonstrate that the shape of the time decay function learned by the convolutional layer can be well approximated by the parametrical function (5). Given that, and for production pipeline efficiency, we use parametrical function (5) as an approximation of *time decay* shape in rest of this paper. But the idea of integrating a convolutional layer into the architecture can be adjusted to more general framework of NN-based recommenders.

Table 1: *Predictor* models with different time-decay

| *Predictor* **models** | **Precision@1** | **PCC@1** |
|---|---|---|
| Default model (3) | 0.072 | 0.018 |
| with conv *time decay* (4) | 0.074 | 0.016 |
| with parametrical *time decay* (5) | 0.075 | 0.017 |

### 3.5 NN BASED RECOMMENDER WITH COMBINED PREDICTOR AND AUTO-ENCODER MODELS

On Figure 3 we observe that *predictor* model has highest precision@K but lower PCC@K. The reason for lower PCC lies in the method of splitting the training data as we train the model to predict more recent purchases from the past week. As a result NN model cannot predict products which

were purchased only one month ago because they were never seen in training output data $Y$. *Predictor* model is recommending personalized popular products. It captures short term preferences and recommend products which are popular now. One of the methods of increasing PCC is to use auto-encoder neural network (Hinton & Salakhutdinov, 2006), which learns representations of the input data X by learning to predict itself in the output. By training the representation and reconstructing all of a user's purchases, it can capture *static* customer preferences and predict products purchased at any time in the past. As a result, we can expect an increase in PCC in *auto-encoder* compared to *predictor* model since the diversity of products over a two year period learned by an auto-encoder is more than a one week period learned by a predictor.

In light of this, we introduce a hybrid approach which combines *predictor* and *auto-encoder* models. We show that they can increase both precision and PCC, or increase one of them and do not change another one.

The combined model may be illustrated by the equations describing the per-layer functions and shown on Figure 2c.

$$A_0 = Td(XT, ti) + \xi; \quad A_j = Drop(Act(W_j \cdot A_{j-1} + B_j)); \quad \hat{Y} = Sigm(W_L \cdot A_{L-1} + B_L);$$

$$loss = Cost(XT, X, \hat{Y}) + \lambda \cdot \sum_{j=1}^{L} |W_j|^2 + \beta \cdot \sum_{j=1}^{L-1} KL(\rho, \rho'_j); \tag{6}$$

$$Cost(XT, X, \hat{Y}) = auto(XT, X, \hat{Y}) + predict(Y, \hat{Y}), \tag{7}$$

in which

$$auto(XT, X, \hat{Y}) = -\sum_{p=0}^{P-1} \{wa_1 \cdot Td(xt_p, to) \cdot log(\hat{y}_p) + wa_0(1 - x_p) \cdot log(1 - \hat{y}_p)\} \tag{8}$$

$$predict(Y, \hat{Y}) = -\sum_{p=0}^{P-1} \{wp_1 \cdot y_p \cdot log(\hat{y}_p) + wp_0 \cdot (1 - y_p) log(1 - \hat{y}_p)\}. \tag{9}$$

Here, we have $wa_1, wa_0$ – constant weights for purchased and non-purchased products respectively, set to be 1 by default. They introduced for balancing number of purchased vs non-purchased products, these weights are used for *auto-encoder* model; $ti$ – time decay parameter applied on input data; $to$ – time decay parameter applied on output data in cost function; $XT$ – purchase dates of products X. $xt_p$ is the scalar value of vector $XT$; $Td(d, decay)$ – time decay function defined in (5) applied on input data and cost function of *auto-encoder*; $d$ – purchase date; $decay$ – time decay hyper-parameter measured in days. The remaining variables are as in model (3).

*Auto-encoder* (8) and *predictor* (9) models are trained jointly using mixed cost function (7). Each output label X in the *auto-encoder* is weighted by the function $Td()$, which is controlled by decay parameter. It controls the impact of *auto-encoder* model. We selected function (5) for controlling the impact of *auto-encoder*. We also apply time decay on the input data $XT$ where shape of time decay function is defined by parametrical definition (5) or can be learned using a conv layer (4). Model (7) can be interpreted as multi-task learning (Caruana, 1998), where loss for predicting past purchases (8) is task one, and loss for predicting future purchases (9) is task two. Model (6) is a generalized version of model (3) and by adjusting $ti$ and $to$ we can get *predictor* model (3) or a mixture of *predictor* and *auto-encoder* models.

*Predictor* model is defined by fixed parameters, $ti = 10^6, to = 10^{-6}$, $wa_1 = wa_0 = 0$ and other NN parameters were found using hyper-parameter optimization on validation data as described in section 3.1. Shape of the time decay function $Td()$ applied on input and output data are shown on Figure 5a by red and black color respectively. *Predictor* model uses past consumptions X as training input data and predicts future purchases $Y$, therefore it has higher precision and lower PCC.

Our first proposed approach is Equally Weighted (*EW*) *auto-encoder* and *predictor* model. It is defined by fixed parameters, $ti = 10^6, to = 10^6$. Because $ti$ and $to$ are very large, there is no time

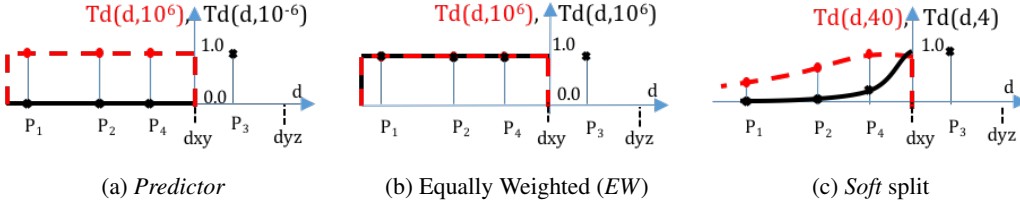

(a) *Predictor*       (b) Equally Weighted (*EW*)       (c) *Soft* split

Figure 5: Different weighted combinations of *predictor* and *auto-encoder* models

decay applied. Shape of the time decay functions applied on input data and cost function (8) is shown on Figure 5b. It uses past purchases $X$ as training input data and predicts both past $X$ and future purchases $Y$. We can expect higher PCC in comparison to *predictor* model because *auto-encoder* and *predictor* model can predict products which belong to both input $X$ and output $Y$ data sets.

Another approach is to use time decay weighted *auto-encoder* and *predictor* model (*soft* split). It is defined by parameters: $ti = 40, to = 4$, which were found using hyper-parameter optimization. It applies time decay on input data $X$ to capture the importance of recent purchases. Its shape is shown in Figure 5c by red color. This model predicts both past and future purchases, but the importance of predicting past purchases is weighted by function $Td()$ highlighted by the black curve in Figure 5c. Therefore, it will allow to keep properties of *predictor* model (for example high precision) and at the same time have properties of EW model (for example, higher PCC). We call this model *soft* split because the training output data has time decayed training input data included in it, compared to the *hard* split of predictor model based on a fixed date split.

## 4 EXPERIMENTS

We evaluate *EW* and *soft* split NN models on AIV dataset. Then we present our model's scalability to work on different digital product categories, and explicate the learning pattern along training process. We also demonstrate the model's ability to capture seasonality effects.

### 4.1 METRICS ON AMAZON INSTANT VIDEO (AIV)

As observed in Figure 3, *predictor* model has high precision against other methods, but comparatively low PCC. We trained both *EW* and *soft* split NN model to check how our proposed model is impacting accuracy metrics.

We select *predictor* model and production collaborative filtering from Figure 3 to serve as baselines, other approaches are excluded because NN is already outperforming. All pre-processed data and model metrics remain unchanged. Results are shown in Figure 6a. We observe that *predictor* model has higher precision and lower PCC than other methods. The model with *soft* split has slightly lower precision, but at the same time PCC is boosted to be twice. *EW* model increases diversity of recommendations by two times, but at the same time precision was reduced significantly. That in return explains our intuition of combing *predictor* and *auto-encoder* models in a *soft* way.

### 4.2 OFFLINE EVALUATION ON PUBLIC DATA SETS

We have evaluated several recommenders on public data sets (MovieLens) and showed that both predictor and soft split models have the highest accuracy metrics in comparison with the existing baseline. More details can be found in the Appendix.

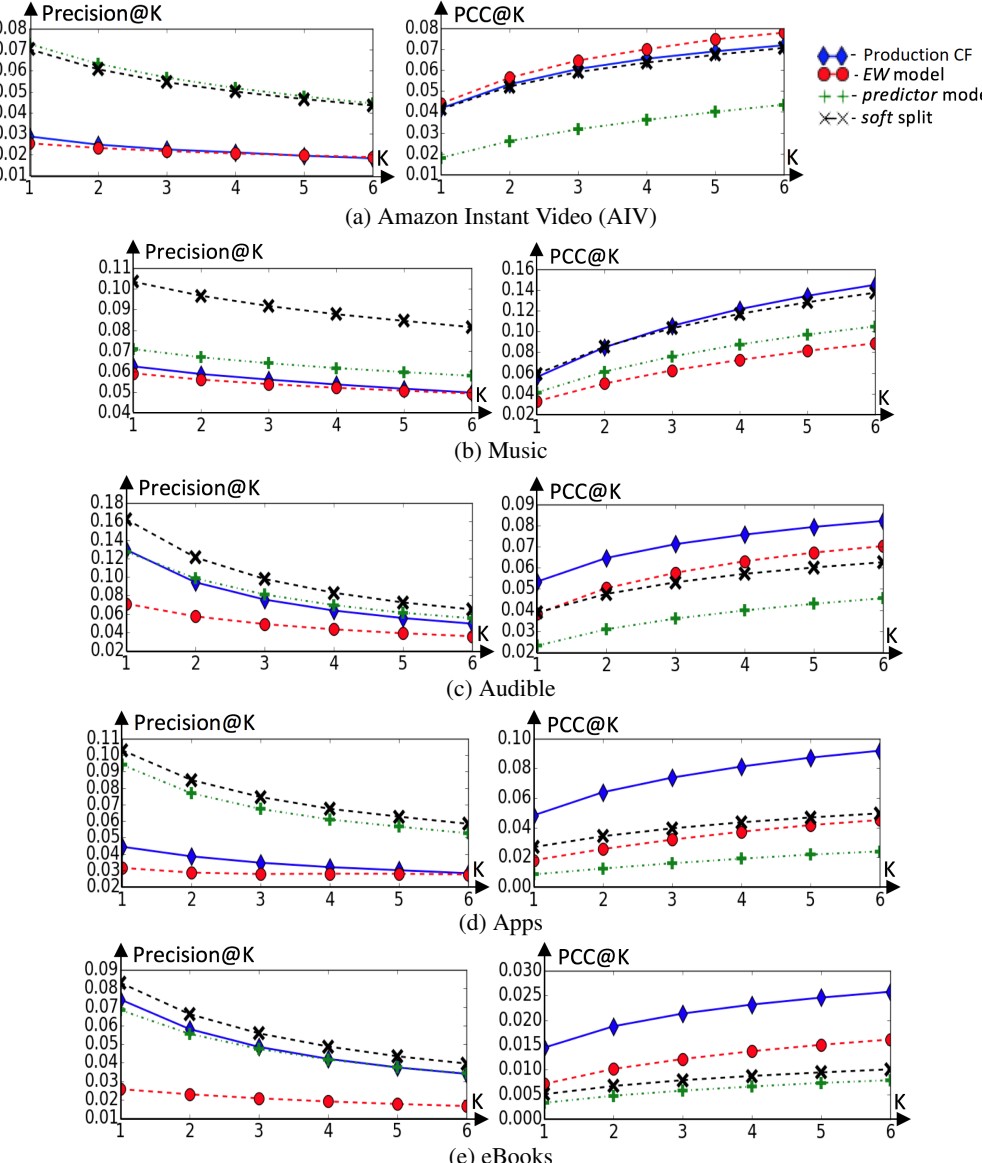

Figure 6: Accuracy metrics on different categories

## 4.3 MODEL SCALABILITY

It is important for recommendation algorithms to be easily operated across different categories, especially at Amazon scale. In this section, we present how our approach can be trained on all categories of digital products without additional feature engineering.

As expected, *auto-encoder*'s ability to model long term static user preference allows us to increase PCC. Model with *soft* data split has higher precision in comparison with other approaches and higher PCC than *predictor* model. But PCC is comparable or lower than *EW* model (Figure 6).

Production collaborative filtering has higher PCC than any other approaches (Figure 6). One of the reasons why NN based recommender has lower PCC is due to input/output dimensionality reduction that we apply in production for daily model re-training.

We show that by applying the learning process of proposed model to different categories dataset, we can gather information from those different patterns. For example, *predictor* approach is modeling short-term user preferences: given two years history of past purchases it predicts future purchases

in the next week. This approach works well on popularity driven categories like video, in which customers are biased more to purchasing popular video products. It has the highest precision@K, but PCC@K is much lower in comparison with other models. *Soft* split allows to keep high precision@K and at the same time increase PCC@K by two times (Figure 6a).

Meanwhile, there are other categories which have different consumption pattern and *predictor* model is not the best. For example, we observe that *predictor* model on video data (Figure 6a) has much higher precision than production collaborative filtering. *Predictor* model is producing personalized recommendations which are popular now. In contrast, on eBooks and audio-books (Figure 6e, 6c) we observe that *predictor* model has the same precision with production collaborative filtering, it indicates that consumption pattern of these two categories is different with video data set (where *predictor* has the highest precision). In both eBooks and Audible model with *soft* split has the highest accuracy, so there is a consumption pattern similarity between both of these categories. *Soft* split model outperforms predictor model on all accuracy metrics on ebooks and audiobooks data sets. On Music and Apps dataset, *soft* split has significantly higher Precision and higher PCC than *predictor* model. We showed that by combining predictor and auto-encoder models we can achieve significant accuracy improvements. The simplicity of the model allows us to scale it in production on all digital categories. It makes it different from other referred papers where one category is picked and then a model is specifically designed for it.

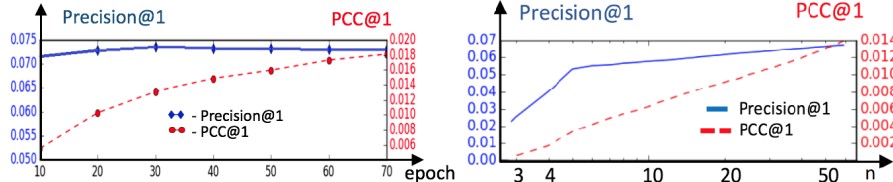

Figure 7: Properties of *predictor* model on AIV data. (a)(left) Convergence of metrics over epochs. (b)(right) Metrics dependent on minimum number of purchases

## 4.4 MODEL PROPERTIES

To understand the strength and weakness of our model, we explore the convergence properties in NN model training process. Here we demonstrate how Precision and PCC change along epochs, together with how they are affected by number of purchases. From Figure 7, we observe that precision converged after 10 epochs, whereas PCC is converging after 70 epochs. NN learns most popular products and reaches high precision fast. With more iterations precision stays stable, but PCC increases significantly as NN learns to predict less popular products or the tail of the products distribution (PCC is increased by 3 times after 70 epochs).

It is observed in Figure 7 that the model accuracy depends on number of purchases customer has. Recommendations will be more popularity biased if customer is inactive and has few purchase, therefore as a result, becomes less accurate. As shown in the figure, Precision@1 and PCC@1 are low on customers who has number of purchases $n <= 3$, but for customer who has a lot of purchases, recommendations will be more accurate and more diverse: Precision@1 and PCC@1 are increasing with growth of minimum number of purchases $n$.

Influence of different number of layers are also studied. We noticed that deeper NN models did not improve accuracy significantly and two hidden layers was enough. This is not surprising as we do not use more complex feature (product description, customer features etc) as in (Covington et al., 2016). Number of hidden units in hidden layer was important for increasing accuracy metrics, thus we did hyper parameters optimization with number of hidden units in range $128, \ldots, 8000$. On the most of the categories, NN with ReLu activation has higher accuracy metrics. This result is consistent with (Covington et al., 2016).

## 4.5 SEASONALITY EVALUATION

To ensure that our model captures seasonality changes or trendiness, we re-train it every day and test how it reacts to seasonal effects. This is validated by comparing the most popular products produced by NN with the most popular products which are actually purchased by customer at different time.

We trained and evaluated NN model on two data sets: October and Christmas data sets, as described earlier in Section 3.4.

In Table 2, we show the most popular products recommended by our model for each date set on left side, and as a contrast, the most purchased ones on the right side. The overlaps between the most recommended and the most purchased products are highlighted in bold. By training the model every day and targeting on future purchases, it successfully captures temporal dynamics. For example, the old movie "National Lampoon's Christmas Vacation" is recommended most during Christmas season and at the same it was the most purchased one. So our model is able capture the trendiness of movies that are watched during Christmas.

Table 2: Top six the most recommended and the most popular videos at different dates

| Most recommended in October | Most purchased in October |
|---|---|
| **Jackass Presents: Bad Grandpa(2013)** | **Jackass Presents: Bad Grandpa(2013)** |
| **Defiance(2009)** | Transformers: Age of Extinction(2014) |
| **The Duchess(2008)** | Space Jam(1996) |
| All Is Lost(2013) | **Defiance(2009)** |
| **Neighbors(2014)** | **The Duchess(2008)** |
| Thanks For Sharing(2013) | **Neighbors(2014)** |
| Most recommended in Christmas | Most purchased in Christmas |
| **National Lampoon's Christmas Vacation(1989)** | **National Lampoon's Christmas Vacation(1989)** |
| **Cirque Du Soleil: Worlds Away(2012)** | **Scrooged(1988)** |
| **Scrooged(1988)** | **Skyfall(2012)** |
| **Skyfall(2012)** | Elysium(2013) |
| The Hunger Games(2012) | The Lone Ranger(2013) |
| Spring Breakers(2013) | **Cirque Du Soleil: Worlds Away(2012)** |

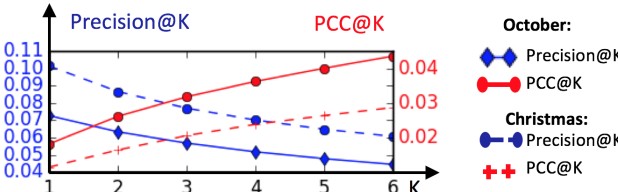

Figure 8: Metrics in October and Christmas

We also measure accuracy metrics to validate the performance of NN models quantitatively. Precision@K and PCC@K are calculated in October and Christmas week and shown on Figure 8. As can be seen, Christmas recommendations has higher Precision@K and lower PCC@K, in comparison with October recommendations. There is an acceptable temporal variation in the accuracy metrics, so we can confirm that out NN model is adapting to seasonality changes. At the time of this test our model was not in production yet, so our recommendations did not impact customer consumption pattern.

In spite of these promising offline results, we must recognize that these algorithms are evaluated against historical data. That data is based on customer consumption history influenced by collaborative filtering algorithms running in production. Therefore, online A/B tests are conducted and discussed in next section.

## 5 RUNNING NN MODELS IN PRODUCTION AND ONLINE A/B TEST

One of the pipeline requirements is to produce fresh NN model with recommendations every day. Another one is to scale this approach to all digital products. For this reason, the designed NN pipeline as shown in Figure 9, consists of several steps: data pre-processing, model training, recommendations generation and serving database of recommendations. It allows us to train multiple NN models for different categories offline, then generate recommendations for all customers offline and store it into database, and in the end serve multiple databases online for all digital products.

We re-train NN model every day and generate recommendations for all customers belonging to a selected category daily. This approach will produce different recommendations every day even if particular customer has not bought anything recently. Daily re-training also helps to learn new items which become popular at that time. Date splits, described in section 3.1, are updated every day so

that $dyz = current\ day$, $dxy = dyz - one\ week$, $dx = dyz - two\ years$. We feed $XY$ data to the input of the NN model and produce output scores $\hat{Y}$. These scores are sorted and the top $K$ products with the highest scores are returned as recommended products. Before sorting, all previous purchases (products belonging to data $XY$) of the selected customer are removed from the recommendations so that only products which customer has never purchased before are recommended.

The first step of the pipeline is data pre-processing, described in Section 3. For this we use Spark on multi host map-reduce cluster. Training time can be impacted by number of training samples, so we control it by keeping number of training samples constant and by removing noisy samples. Another factor impacting training time is the model size which is defined by number of recommended products and number of units in hidden layer. We control number of recommended products by clipping the tail of the product distribution. The next step is neural network model training. Training time is governed by the size of the weight matrixes $W_j$. Bigger size of model takes more time to train. We use multi-GPU model parallelization which we implemented in the DSSTNE library (10) and open-sourced it. Model parallelization allows us to train a neural networks with million of input and output dimensionalities (so that model size can be more than several gigabytes) in timely manner.

In order to scale up and compute the predictions daily we do data parallel predictions. We split data into batches of customers and run predictions through Spark on a GPU cluster (11). Each predictor receives customer purchase history and the trained NN model generates recommendations (predictions) which are stored in database. The online recommender service reads these predictions and displays them to customers.

Above steps are repeated every day, so that NN model uses the freshest purchase history and can capture seasonality changes.

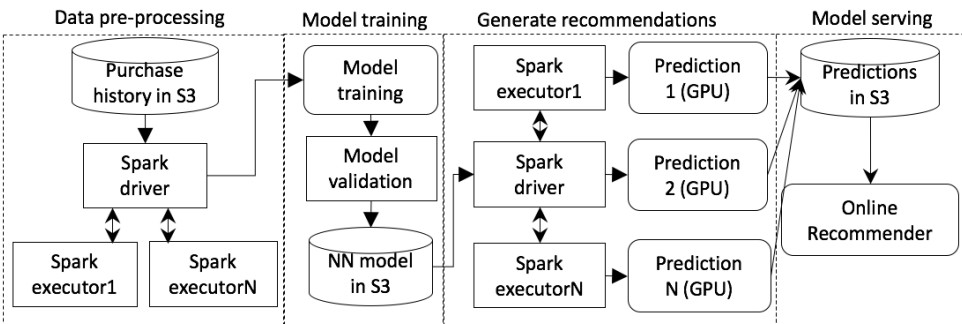

Figure 9: Production pipeline

Our pipeline is similar with (Cheng et al., 2016), the main difference is that we pre-generate recommendations offline. As a result, our online recommender remains simple and supports multiple categories with no modifications. Also Data pre-processing, Model training and Recommendations generations pipeline can be used by multiple category recommenders sequentially or in parallel if there is an extra hardware available. The side effect of our approach is offline recommendations generation is delaying the propagation of recent purchases into our model.

We conduct online A/B tests of the proposed NN based recommender on several digital categories. The length of the test is four weeks. Fifty percent of all customers in the selected category are exposed to new recommendations and remaining fifty percent to existing recommendations. We observed p-value to be less than 0.05 for all categories and significant improvements in number of purchases of mobile apps on Apps Storefront (Amazon digital devices), ebooks on Kindle devices, and audio books on Audible. We also observed a similar increase in Amazon Instant Video minutes streamed on Amazon.com.

## 6  CONCLUSION AND FUTURE PLANS

We described a personalized neural network based recommender system which was launched in production on categories like eBooks, Audible, Apps and Video. We are currently working on expand-

ing it to non-digital categories. We showed that splitting customer purchases into a history period (input) and a future period (output) in our models led to good results, and some of our production models use this approach (with *soft* split which combines the *auto-encoder* model with the future *predictor* model). We have applied *time decay* learnt by convolutional layer, or defined by parametrical function to consumption event. It captures the importance of recent activity, and combined with *soft* split, it leads to significant improvements in offline metrics. We demonstrated that two layer neural networks are outperforming other NN based approaches which are more complicated than our method, both on public dataset (MovieLens) and company's internal datasets. Because of simplicity of the NN model we can scale it on all digital categories. We observed significant KPI improvements during online A/B tests in production. Finally we open sourced our deep learning library which supports multi gpu model parallel training and allows us to train large models in timely manner.

## ACKNOWLEDGMENTS

We would like to thank Charles Elkan, Ainur Yessenalina and Zach Lipton for their help with ReLU neural network and proposing the products converted coverage metric. We would also like to thank Mihir Bhanot, Vishnu Narayanan, Liping Zhou, Vijay Garla, Varun Jagannath, Wim Verleyen, and Audible and eBooks teams for running the A/B tests. We would like to thank Shaun McCarthy and his team for sharing video data and insightful discussions. We would also like to thank Vishy Vishwanathan and Choon Hui Teo for their valuable comments. We would like to thank Srikanth Thirumalai, Amber Roy Chowdhury, Matias Benitez and everybody on the ReMo team for their support on recommendations modeling.

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

## A    OFFLINE EVALUATION ON MOVIELENS DATA SETS

We compare our models (predictor, soft split) with FastXML based recommender on 20M Movie-Lens data set (Harper & Konstan, 2015). FastXML is selected for baseline accuracy estimation because it is designed for extreme multi-label problem. It is optimizing ranking loss which is important feature for recommender problems, and it has the best accuracy among other baseline methods on Figure 3.

MovieLens data is selected because it belongs to the same category (video) which was evaluated earlier in this paper. Our recommendation approach is based on positive implicit feedback, we chose ratings greater than three as relevant for the user and ignore all the other ratings (for consistency we will call all ratings greater than three as purchases). Similar conversion of the ratings to implicit feedbacks was used in (Ostuni et al., 2013).

We train model to predict future purchases, so we split training data into past ($X$) and future ($Y$) purchases using dates: $dx$ = 2002/04/14; $dxy$ = 2004/04/01; $dyz$ = 2004/04/14; $dz$ = 2004/04/28. We generate training input and output data by selecting users that have at least two purchases in period of time $dx \ldots dxy$ and at least one purchase in period of time $dxy \ldots dyz$. So we get 923 customers in training data. Purchases belonging to dates $dx \ldots dxy$ assigned to training input data $X$, and belonging to $dxy \ldots dyz$ assigned to training output data $Y$.

There are around 6200 products (movies) purchased (rated) by these customers in the above period of time. Distribution of customers sorted by number of purchases is shown on Figure 10 (b), where $H(c)$ number of purchases made by customer $c$, $c$ customer index. It shows that 90 percent of the customers have less than 400 purchases. Distribution of products in data $X$ and $Y$ are shown on Figure 10 (a), where $PX(p)$, $PY(p)$ number of purchases of product $p$ in the input ($X$) and output ($Y$) training data accordingly, $p$ product index. Both of these distributions have long *tail*: 90 percent of purchases are covered by 1000 products.

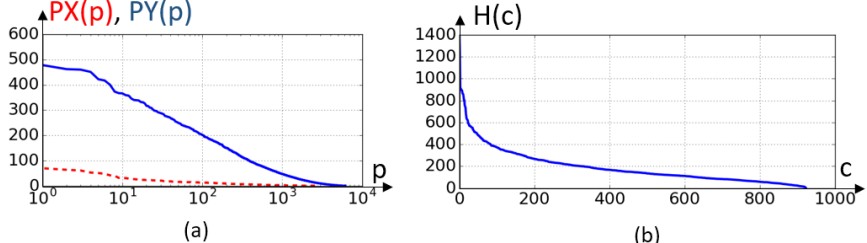

Figure 10: Distribution of products (a) and customers (b)

During evaluation we feed $XY$ data to the models and produce output scores $\hat{Y}$. These scores are sorted and the top K products are returned as recommendations. Before sorting, all previous purchases (products belonging to data $XY$) of the selected customer are removed from the recommendations, so that only new products are recommended. These recommendations are compared with future purchases (data $Z$) for accuracy calculation. We get testing input data XY and testing output data Z by selecting customers who has at least two purchases in period of time $dx \ldots dyz$ and at least one purchase in period of time $dyz \ldots dz$. There are 921 customers who satisfy these conditions. Purchases belonging to dates $dx \ldots dyz$ assigned to testing input data $XY$, and belonging to $dyz \ldots dz$ assigned to testing output data $Z$. Accuracy metrics of predictor, *soft* split and fastXML models are presented on Figure 11.

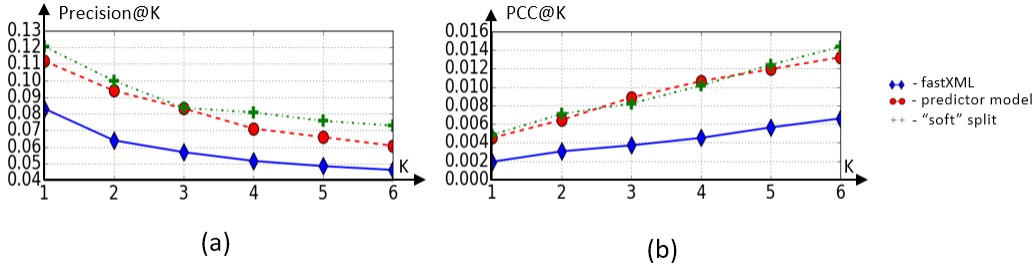

Figure 11: Accuracy metrics

Predictor model has the same PCC@K with *soft* split and lower precision@K. Both of these models have higher accuracy metrics than fastXML. We observe similar difference in precision between predictor model and fastXML on Figure. 3, but fastXML has higher PCC@K. These results can be used only as an approximation of a performance on real implicit feedbacks (purchase history), because in this section we were using ratings converted to implicit feedbacks.

