# OpenReview forum: "THE EFFECTIVENESS OF A TWO-LAYER NEURAL NETWORK FOR RECOMMENDATIONS"
_ICLR.cc/2018/Conference — Invite to Workshop Track_

### Official Review · AnonReviewer1 · 2017-11-17
**Good industrial/empirical paper with some surprising findings.**

**Rating:** 6
**Confidence:** 3

**Review:**

The paper proposes a new neural network based method for recommendation.

The main finding of the paper is that a relatively simple method works for recommendation, compared to other methods based on neural networks that have been recently proposed.

This contribution is not bad for an empirical paper. There's certainly not that much here that's groundbreaking methodologically, though it's certainly nice to know that a simple and scalable method works.

There's not much detail about the data (it is after all an industrial paper). It would certainly be helpful to know how well the proposed method performs on a few standard recommender systems benchmark datasets (compared to the same baselines), in order to get a sense as to whether the improvement is actually due to having a better model, versus being due to some unique attributes of this particular industrial dataset under consideration. As it is, I am a little concerned that this may be a method that happens to work well for the types of data the authors are considering but may not work elsewhere.

Other than that, it's nice to see an evaluation on real production data, and it's nice that the authors have provided enough info that the method should be (more or less) reproducible. There's some slight concern that maybe this paper would be better for the industry track of some conference, given that it's focused on an empirical evaluation rather than really making much of a methodological contribution. Again, this could be somewhat alleviated by evaluating on some standard and reproducible benchmarks.

---

> ### Author Response · Authors · 2017-12-06
> **Review Rebuttal for Reviewer1 about benchmarking on public data sets and methodological contribution**
>
> Thank you for your feedback.
> 1. Comments about benchmarking on public data sets:
> You raised a good point about evaluation on public data sets. It was not done because of several reasons:
> 1.1 There is no public data sets which have the same properties with our data: implicit feedbacks(purchase events + date of purchase), large number of products, large number of customers.
> 1.2 Most of the papers are reporting RMSE or precision@K on randomly held out data sets, whereas we measure precision@K at particular time (future week). So, that estimated metrics are as close as possible to real production environment.
>
> We would like to alleviate your concern about evaluation on public data sets.
> We are going to pick MovieLens data [http://files.grouplens.org/datasets/movielens/ml-20m.zip] because it is related to one of the categories we use in the paper.
> We are going to convert all rating to watches events (implicit feedbacks) by thresholding the ratings: 1 if rating >= 3, 0 otherwise. The same implicit feedback conversion was used in the paper [Vito Claudio Ostuni et. Al Top-N recommendations from implicit feedback leveraging linked open data. RecSys '13]
> We are going to split the MovieLens data into past and future purchases. Then use past purchase for training the models and future for evaluation.
> In the end we will compare accuracy metrics of our method with existing techniques on MovieLens data sets and report precision@K and PCC@K on future week (as described in point 1.2).
> Please let me know if above approach can alleviate your concern about bench-marking on public data sets.
>
> 2. Comments about our contribution:
> 2.1 Yes, one of the focus of this paper is scaling neural network based recommender on all digital categories in real production environment.
> We also open sourced the core library which is used in our experiments. It supports multi-gpu model parallelization. It allows us to train a neural networks with million of input and output dimensionalities (so that model size can be more than several gigabytes) in timely manner.
>
> 2.2 In addition to that we did methodological contribution (which was a key for success in running these models in production):
> We proposed to use convolutional layer for exploring the shape of the time decay.
> We proposed different methods for increasing precision@K and PCC@K of neural network based recommender. We presented results on different data sets such as video, audiobooks, ebooks, music
> These data sets have different properties:
> a). For example on video data sets (which are popularity biased) we showed that predictor model combined with time decay on input data improves precision@K only, we also showed how to combine predictor model with auto-encoder so that PCC@K can be increased 2 times without significant reduction of precision@K.
> b). On other data sets like ebooks and audio-books (which are less popularity biased then video data) we showed that combination of predictor and autoencoder models increases both precision@K and PCC@K in comparison with predictor model.
> The goal of this project was to find a solution which can be scaled to all digital categories of retail catalog. It makes it different with other referred papers where one category is picked and then model is specifically designed for it.
>
> We will add above comments in the next revision of the paper.

---

### Official Review · AnonReviewer3 · 2017-11-26
**Good use of known techniques to build a production ready recommender model**

**Rating:** 6
**Confidence:** 4

**Review:**

Authors describe a procedure of building their production recommender system from scratch, begining with formulating the recommendation problem, label data formation, model construction and learning. They use several different evaluation techniques to show how successful their model is (offline metrics, A/B test results, etc.)

Most of the originality comes from integrating time decay of purchases into the learning framework. Rest of presented work is more or less standard.

Paper may be useful to practitioners who are looking to implement something like this in production.

---

> ### Author Response · Authors · 2017-12-13
> **Review Rebuttal for Reviewer3**
>
> Thank you for your feedback
> In addition to your comments we would like to highlight several points:
> 1. Two methods of integrating time decay of purchases into the learning framework were proposed:
> 1.1 Convolutional layer for exploring the shape of the time decay function.
> 1.2 We explored properties of different neural network based recommenders: predictor and auto-encoder models and proposed a method of combining their properties by integrating time decay of purchases into the learning framework. The final model (“soft” split) can be interpreted as a generalized auto-encoder which has time decay on input layer and time decay on cost function. The evaluation is done on both internal and public data sets.
> 2 Our relatively simple model can capture seasonality changes with daily re-training.
> 3 We designed and open-sourced the core library which was used on these experiments. This library supports multi-gpu model parallel training and allows us to train large neural networks based recommender (model size can be more than several GB) in timely manner.
> 4 Our approach is successfully scaled and outperforms existing recommenders on four different categories of one of the largest retail catalog.
>
> Several reasons of emphasizing the online production A/B test results are presented below:
> I would like to highlight the importance of reporting the online A/B test results for recommender systems which was done in this paper. The standard evaluation of real recommender system estimates KPI gain (for example number of purchases) and confidence level (p-value).
> If p-value is low and KPI gain is high it means that we are confident that KPI gain has low probability of being random.
> If p-value is high it means that we are not confident in results and it is highly probable that KPI gain is random.
>
> That is why if  only offline metrics improvements of recommender system with no confidence evaluation is reported, then we do not know what is the probability of the offline gain being random. For example we report offline improvements on next week, but how about second, third week and etc. Even if we produce the full accuracy distribution over next several months it will not be real because of the second point below.
> Second point about "pure" offline evaluation: it is done on purchases made by customers which were exposed to recommendations produced by different recommender (for example by legacy recommender). So again, offline metrics do not show real picture. In this case even if there is no gain in offline metrics we still can get KPI gain during online test and vice versa.
> During online A/B test the recommender loop is “closed”: we are evaluating KPI metrics on purchases which were done by customers who are exposed to the recommender which we are evaluating.
>
> In conclusion offline evaluation is a preliminary test of the opportunity (which says that designed method can produce some recommendations) and only online A/B test shows real value of the designed approach. That is why we highlighted the last point in our paper: we demonstrated low p-value (less than 0.05) and increased KPI (number of purchases).

---

### Official Review · AnonReviewer2 · 2017-11-30
**A good applied paper with a novel approach and good experimental results**

**Rating:** 7
**Confidence:** 3

**Review:**

This paper presents a practical methodology to use neural network for recommending products to users based on their past purchase history. The model contains three components: a predictor model which is essentially a RNN-style model to capture near-term user interests, a time-decay function which serves as a way to decay the input based on when the purchase happened, and an auto-encoder component which makes sure the user's past purchase history get fully utilized, with the consideration of time decay. And the paper showed the combination of the three performs the best in terms of precision@K and PCC@K, and also with good scalability. It also showed good online A/B test performance, which indicates that this approach has been tested in real world.

Two small concerns:
1. In Section 3.3. I am not fully sure why the proposed predictor model is able to win over LSTM. As LSTM tends to mitigate the vanishing gradient problem which most likely would exist in the predictor model. Some insights might be useful there.
2. The title of this paper is weird. Suggest to rephrase "unreasonable" to something more positive.

---

> ### Author Response · Authors · 2017-12-06
> **Review Rebuttal for Reviewer2 about vanishing gradient, explanation of lower performance of LSTM and paper title**
>
> Thank you for your feedback.
>
> 1. Comments about vanishing gradient:
> We acknowledged that we did not add detailed info about vanishing gradient of predictor model(feedforward neural network). More details with experimental results are presented below:
> We use ReLU activation function to mitigate vanishing gradient in predictor model.
> With increase of the depth (number of hidden layer) of predictor model, accuracy metrics can degrade significantly (vanishing gradient is one of the reason of such effect). That is why we measured the impact of the NN depth on Precision@1, and observed that with increasing the NN depth, Precision@1 is going down as follow (even with ReLU):
> Depth                1            2        3         4         5         6
> Precision@1   0.072   0.072   0.07   0.068  0.067  0.065
> One of the method of mitigating the accuracy degradation (due to depth of NN) is residual neural networks [K. He, X. Zhang, S. Ren, and J. Sun. Deep residual learning for image recognition. In CVPR, 2016.]. We explored residual NN with predictor model on our data sets, and observed that it mitigates vanishing gradient effect, so that precisoion@1 stayed the same regardless of the depth of the neural network: around 0.072. But it does not improve accuracy metrics in comparison with two layers NN. That is why we picked neural network model with number of hidden layer no more than 2.
> We will add these comments with experimental results in the new paper revision.
>
> 2. Comments about LSTM performance:
> LSTM is well applied on sequences like text, speech etc. These sequences has “strong” grammatical rules, which are well captured by LSTM. We explain lower accuracy of LSTM by our data properties (or lack of “strong” grammatical rules in sequences of purchases in our data). For example on ebooks data, if one customer buy books in order: “Harry Potter”, “Golden Compass”, “Inkheart”, another customer can buy these books in different order:  “Inkheart”, “Harry Potter”, “Golden Compass” and another one in different order, etc. So these purchases can be in any order and “long” term dependencies can be noisy.
> Another important properties of our data(video, ebooks) is the popularity of the recommended products at particular date. Our approach (predictor model) is modeling these properties by re-training the model every day and predicting the next purchases which are popular in the current week, whereas LSTM is recommending only next purchases (which are not necessary popular at current week).
> We can expect better performance of LSTM on other categories of products (where order of purchases is more important), for example probability of buying a game for a cell phone after purchasing a cell phone is higher than probability of buying these products in reversed order.
>
> 3. Comment about paper title:
> We will rename it to:  “THE EFFECTIVENESS OF A TWO-LAYER NEURAL NETWORK FOR RECOMMENDATIONS”
>
> We will add above comments in the next paper revision.

---

### Author Response · Authors · 2017-12-13
**Thank you to reviewers**

We would like to thank all the reviewers for their careful consideration of our paper, and very useful comments.
We provided detailed responses below and uploaded a new revision of the paper

---

### Decision · Program_Chairs · 2018-01-29
**ICLR 2018 Conference Acceptance Decision**

**Decision:**

Invite to Workshop Track

**Comment:**

Meta score: 6

This is a thorough empirical paper, demonstrating the effectiveness of a relatively simple model for recommendations:

Pros:
 - strong experiments
 - always good to see simple models pushed to perform well
 - presumably of interest to practioners in the area

Cons:
 - quite oriented to the recommendation application
 - technical novelty is in the experimental evaluation rather than any new techniques

On balance I recommend the paper is invited to the workshop.